# A Comparative Analysis of Implants Presenting Different Diameters: Extra-Narrow, Narrow and Conventional

**DOI:** 10.3390/ma13081888

**Published:** 2020-04-17

**Authors:** Henrique Tuzzolo Neto, Alessandra Sayuri Tuzita, Sérgio Alexandre Gehrke, Renata de Vasconcellos Moura, Márcio Zaffalon Casati, Alfredo Mikail Melo Mesquita

**Affiliations:** 1Dental School, Universidade Paulista—UNIP, São Paulo 04026-002, Brazil; henriquetuzzolo@hotmail.com (H.T.N.); renata_vmoura@hotmail.com (R.d.V.M.); mzcasati@gmail.com (M.Z.C.); alfmikail@yahoo.com.br (A.M.M.M.); 2Biotechnology Department, Universidad Católica de Murcia, 11.100 Murci, Spain; sergio.gehrke@hotmail.com

**Keywords:** narrow implants, extra-narrow implants, diameters of implants, dental implants

## Abstract

This study aimed at performing a comparative analysis of the fracture resistance of implants, evaluating extra-narrow, narrow, and regular implants. Four groups containing 15 implants each were evaluated. Group 1 (G1): single-piece extra-narrow implants; Group 2 (G2): single-piece narrow implants; Group 3 (G3): Morse taper narrow implants with solid abutments; Group 4 (G4): Morse taper conventional implants with solid abutments. The implants were tested using a universal testing machine for their maximum force limit and their maximum bending moment. After obtaining the data, the Shapiro–Wilk, ANOVA, and Tukey (*p* < 0.05) statistical tests were applied. Samples from all the groups were analyzed by scanning electron microscopy and Groups 3 and 4 were analyzed by profilometry. The means and the standard deviation values for the maximum force limit (N) and the maximum bending moment (Nmm) were respectively: G1:134.29 N (10.27); G2:300.61 N (24.26); G3:360.64 N (23.34); G4:419.10 N (18.87); G1:1612.02 Nmm (100.6); G2:2945 Nmm (237.97); G3:3530.38 Nmm (228.75); G4:4096.7 Nmm (182.73). The groups behaved statistically different from each other, showing that the smallest diameter implants provided less fracture resistance, both in the tensile strength tests and in the maximum bending moment between all groups. Furthermore, single-piece implants, with 2.5 mm and 3.0 mm diameters, deformed in the implant body region area, rather than in the abutment region.

## 1. Introduction

Insufficient space between adjacent teeth and narrow alveolar ridges are frequent findings in an implantologist’s daily practice. Those clinical situations can make rehabilitation with regular diameter (3.75–4.1 mm) implants unviable [1]. As a result, the use of implants with reduced diameters, less than 3.75 mm, has significantly contributed to the restoration of areas with limited prosthetic space [2,3,4,5,6].

However, rehabilitation in areas such as the upper lateral and lower incisors can be very challenging, even when using narrow implants. To manage different clinical scenarios, manufacturers have started producing implants with different diameters. Al-Johany and colleagues [7] have suggested a classification system for dental implants according to their diameter: (1) extra-narrow implants, with a diameter of less than 3.0 mm; (2) narrow implants, with a diameter of 3.0 mm or more, but less than 3.75 mm; and (3) regular implants, with a diameter of 3.75–4.0 mm.

Many factors can cause fracture in implants such as metal fatigue, the design of the prosthetic structure, occlusion forces, bone resorption, and implant diameter [2,6,8,9,10,11,12,13,14]. Narrow implants, when compared to regular implants, present an increased risk of fracture due to their smaller diameter, which may compromise not only the components of the prosthesis but also lead to bone overload [8].

Finite element studies on the influence of the implant diameter have shown that, when using narrow implants (<3.3 mm), the stress is much higher on the abutment region; moreover, the implant diameter significantly affects the strength of the implants [15].

The main aim of this research was to perform a comparative analysis of the fracture resistance of implants, having as a basis the ISO 14801 standards, by evaluating single-piece extra-narrow implants, single-piece narrow implants, Morse taper narrow implants, and Morse taper regular implants, all of them featuring abutments with the same height.

## 2. Materials and Methods

Sixty implants manufactured and commercialized by Implacil de Bortoli (Sao Paulo, Brazil) were divided into four groups of 15 implants each, as shown in Figure 1, as determined by the sample calculation of the pilot study performed before this experimental work.

The implants were vertically embedded in a self-cured methyl methacrylate-based resin (Pattern Resin^TM^ LS GC America INC., Alsip, IL, USA) and placed into a 20 mm diameter polyvinyl chloride (PVC) tube, with 3 mm of its body above the base and with a standardized angulation of 90° to the base, using a parallelogram (PRO-DELL Ltda—Standard Model, Sao Paulo, Brazil).

For Morse taper implants (Group 3 and Group 4), a 6 mm height solid abutment was inserted (Figure 2) with a torque of 30 Newtons, according to the manufacturer’s recommendations.

In order to perform the mechanical test, the samples were tested in a device positioned at 30° ± 2°, simulating the incisors’ area. A ball-shaped metal device, simulating a dental crown, with an opening access for the retaining screw, was cemented on the abutment, enabling a single load point and following the ISO 14801:2007 standard [16] (Figure 3).

Load was progressively applied to the universal testing machine (Kratos, KE series, Kratos-Equipamentos Industriais Ltda, Cotia, Sao Paulo, Brazil) with a speed of 1 mm per minute and pausing when reaching a 6 mm displacement [8].

The hemispherical loading surface and the surface of the loading device were visually inspected after each test to ensure that no deformation occurred.

After applying the test, a graph was generated showing the load/displacement curve of each implant of the superimposed group, and a table was generated showing the data that were obtained (mean and maximum loads). To calculate the maximum bending moment, an equation was applied according to the ISO 14801: M = y.F, in which the resulting unit was given in Newton x millimeter (Nmm).

y = distance from the center of force applied to the intersection point, between the long axis and the upper point of the device.

F = force applied to the tested set.

The characteristics of the load/displacement curve for each implant design were generated, and the maximum (mean) load, the maximum bending moment, and the appropriate standard deviation values were established.

After applying the test, three samples from each group were prepared for SEM. A longitudinal section at the center of each specimen was cut by a metallographic cutter (Isometi 1000). After being cut, the samples were polished using a sequence of abrasive paper (240, 320, 400, 600, and 1200 abrasive grit). Subsequently, the samples were cleaned in an ultrasonic tank with isopropanol [17]. The implants’ characteristics and the abutment/implant specimens were evaluated by scanning electron microscopy (JSM-LV 6510, JEOL, Tokyo, Japan), which provided a descriptive analysis [18].

For illustrative purposes, three samples from Groups 3 and 4, containing two parts each (implant/abutment), were submitted to analysis on a profilometer for better visualization of the distortion on the implant platform. The results were submitted to the Shapiro–Wilk test and, subsequently, to the ANOVA One Way test and, later, to the Tukey test.

## 3. Results

### 3.1. Descriptive Statistics and Data Analysis

The descriptive statistics are presented in Table 1, showing the mean and the standard deviation values of maximum force, tensile strength, and maximum bending moment of each group. 

Therefore, the ANOVA One Way test was performed, presenting a statistically significant difference for all the variables and all the groups regarding the maximum force (N) and the maximum bending moment (N/mm). After performing the ANOVA test and verifying the existence of a statistically significant difference, the Tukey test was applied with *p* < 0.05 for the force and the maximum bending moment, showing a statistically significant difference for all the groups, asrepresented by the letters in the columns (Table 1).

### 3.2. Scanning Electron Microscopy (SEM)

SEM results are shown below, with 12× and 45× magnification (Figure 4, Figure 5, Figure 6 and Figure 7).

Figure 4, referring to Group G1, shows a single-piece implant with a 2.5 mm diameter, with bending in the implant body area, but not affecting the ring and the abutment areas.

Figure 5, referring to Group G2, shows a single-piece implant with a 3.0 mm diameter, with bending in the implant body area, but not affecting the ring and the abutment areas.

Figure 6 presents a 3.5 mm narrow diameter implant and solid abutment, showing that distortion was more concentrated in the implant/abutment interface area. However, there was also distortion in the initial third of the connection, causing opening at the interface side.

Figure 7 presents a 4.0 mm regular diameter implant and solid abutment, showing that distortion was concentrated in the implant/abutment interface area, with larger distortion in the component.

### 3.3. Profilometer

Three samples from Groups 3 and 4 were placed in a profilometer after the abutment removal, aiming at observing the location and the characteristics of the distortions (Figure 8a,b):

The implants’ illustrative images (Groups 3 and 4) superimposed on the profile gauge show a distortion on the implant platform located on one of the walls, where there was a compression of the abutment on the implant, as seen in the SEM.

## 4. Discussion

The comparison between groups showed a statistically significant difference. The larger the diameter, the greater the fracture resistance.

Nowadays, it is very hard to compare dental implants due to the multiple designs available (macro and micro), different types of connections, different chemical compositions, diameters, and sizes, among other differences. The ISO standard for the comparison of implants has several limitations regarding the oral environment because it does not consider the presence of bruxism, the region in which the implant was installed (incisors, premolars, and molars), occlusal load, and other factors. However, it is important to highlight the standardization of laboratory results, which allows the development and evolution of dental materials.

The specification of a breaking point is unclear, according to ISO 14801. As a result, based on the studies using this ISO, the device of the loading hemispheric component recorded the data after 6 mm of displacement. This ISO simulates the worst-case scenario because this normative use oblique load is more critical than axial load, and also because of the fact that the implant pillar junction is kept out of acrylic resin for the purpose of verifying its biomechanical behavior [19,20,21,22].

The fracture of implants by fatigue is one of the main causes of mechanical failure of implants; however, many studies do not demonstrate the relationship between the mechanical reason for failure and the mode of the fracture [4,6,11,12,13,14,18].

Most extra-narrow implants have variable diameters according to the manufacturers, such as 1.8 mm, 2.4 mm and 2.5 mm; however, there is insufficient scientific evidence regarding their success rate. They are indicated for use only in edentulous regions with lesser occlusal load, such as incisors. This condition does not occur with narrow implants with a diameter between 3.0 and 3.25 mm, as they are already indicated for all regions [4].

In a clinical, radiographic, multicenter five-year follow-up on the rehabilitation of upper lateral incisors with 97 two-piece implants of 3 mm diameter, Galindo Moreno et al. [23] did not report changes in bone or gingival level, but reported that the main cause of failure was due to the fracture of pillars.

In this study, in Group 1, single-piece extra-narrow implants of 2.5 mm were used and presented maximum strength values of 134.29 N, which means an almost 50% lower value when compared to Group 2, which used 3 mm diameter single-piece implants with a similar design. This increase in diameter occurs precisely where a larger deflection area occurs, as shown by microscopy, that is, in the region of the pillar/implant interface, which justifies the increase.

The other groups showed mean values of 300.61 N (Group 2); 360.24 N (Group 3) and 419.10 N (Group 4). Starting from Group 2, there was an increase of 0.5 mm in diameter per group, increasing the maximum strength values by 20% on average, regardless of whether the implants were single-piece or two-piece.

In this research, all samples were titanium F 67 (normative that regulates the production of Pure Titanium, for surgical implants) grade 4, which has an elasticity module of 107 GPa, flow limit of 170 MPa and mechanical resistance of 240 MPa [24]. Some companies use modified titanium alloys, with aluminum and vanadium in the composition of the implant, aiming to improve the strength of these devices [15]. The variables involved in implant behavior are difficult to control since the masticatory forces are not constant, and the properties of the material may be different throughout the implant [25].

According to Santos et al. [26], the model and material used for the manufacturing implant-supported components clearly influence the processes of plastic deformation, wear, or failure of prosthetic parts. In the present study, there was no such relationship between the model, material, and failure, considering that the types of pillars tested were of the same material and model and behaved in a similar way in relation to the resistance to loading and the location of distortion.

The identification of the exact starting point of deflection would be of clinical interest, but it is only possible to identify the maximum force. In the methodology applied for Groups 3 and 4, the SEM shows the damage caused by the flexural test, with distortion on the implant platform being more prominent in the pillar/implant interface, showing compression on one side of the implant platform, with the creation of a gap on the opposite side. This fact suggests that the locking system by the imbrication of the internal connection of the Morse taper implants extends the contact of the pillar with the internal walls of the implant, protecting the narrow implants [27,28].

In the implants of Groups 1 and 2, the deflection presented in the microscopy was observed in the same region of the implant, not leading to fracture nor to the appearance of a gap because they were single-piece, but leading to the same implant deformation.

There were distortions in the cervical region of the implants caused by the intermediate pressure provided by the load, besides the presence of an angulation between the implant/abutment interface susceptible to flexural strength. These distortions reached the outer surface, indicating a structural failure of the implant. Previous research from Bordin et al. [29], Freitas et al. [27] and Prados-Privado et al. [30] showed that the comparison between internal cone implants, external hexagon, and internal hexagon showed a clear advantage for internal conical implants in relation to the others, considering the fatigue of the material.

In vitro studies by Gherlone et al. [31] and Gastaldi et al. [32] compared double cone implants with Morse taper internal connection with Morse taper external hexagon implants, showing their resistance against bacterial microinfiltration. This type of connection, when used in immediate loading and late post-extraction implant rehabilitation, showed a good clinical result in 24 months of follow-up.

Internal connections offer advantages such as a reduced vertical height platform for restorative components, the distribution of lateral forces within the implant body, pillar screw protection, and long internal walls that resist detachment of the interfaces [29]. However, in this study, it was observed in the SEM that gaps occurred on the interfaces of Groups 3 and 4, supporting the study by Bordin et al. [33], in which the survival and probability of failure of narrow implants with different diameters were evaluated by comparing 42 implants, presenting the same macrogeometry and internal conical connection, through finite elements and mechanical tests. These implants were divided into two groups according to the diameter (narrow—Ø3.3 × 10 mm and extra-narrow—Ø2.9 × 10 mm). It was observed that there was no significant difference between narrow and extra-narrow implants because the failure mode was similar in both groups, restricted to the fracture component. Also, according to Bordin et al. [29], the intermediate fracture was reported as a prosthetic failure for narrow two-piece implants in the posterior region, and its survival rate was significantly reduced.

Group 1 was the only one that demonstrated fragility at the same intensity in the pillar as in the implant body. Therefore, it is appropriate to say that there may be an increased bone resorption around extra-narrow implants, according to the results of the systematic review performed by Klen, Schiegnitz and Al-Nawas [4], in which they showed that extra-narrow implants (<3 mm) showed an increase in bone loss, attributed to the masticatory load, when compared to conventional narrow implants (3.0 mm to 3.5 mm). Thus, the authors reported that extra-narrow implants would be indicated only for toothless patients in the anterior region.

According to the study of Hirata et al. [34], narrow implants are indicated for areas of limited bone width or when a bone graft is not feasible; however, the reduction of the implant diameter may compromise its mechanical performance, which corroborates the results of this study. For two-piece implants, one of the most reported mechanical problems is the loosening and/or fracture of the pillar screw, in which a fracture may occur or not. In the case of narrow implants, the load and tension will be distributed in a significantly smaller area compared to a regular diameter implant.

In a systematic review conducted by Assaf et al. [35], it has been reported that the use of narrow implants, with a diameter less than 3.5 mm, shows a success rate equivalent to the regular diameter implants, thus being indicated for dental regions with narrow clinical crowns and/or limited spaces. In addition to the implant diameter, the survival of the implant depends on factors such as the implant surface and length, the bone quality and quantity, the absence of peri-implant disease, favorable occlusion, adequate patient hygiene, and professional experience.

Logically, this research conducted in a laboratory does not reflect the dynamic aspects of chewing, occlusal load, type of prosthetic crown material, prosthesis type, bone density, and others, and therefore remains very distant from clinical reality. However, based on the results and what was discussed in this research, the clinical use of single-piece extra-narrow implants with careful indications is recommended, especially in patients with high occlusal risk, such as those with an absence of simultaneous bilateral contacts or occlusal overload by bruxism.

## 5. Conclusions

The groups analyzed behaved differently from one another, showing that smaller diameter implants provided lower fracture resistance, with a statistically significant difference regarding the tensile strength and the maximum bending moment between all the groups.

## Figures and Tables

**Figure 1 materials-13-01888-f001:**
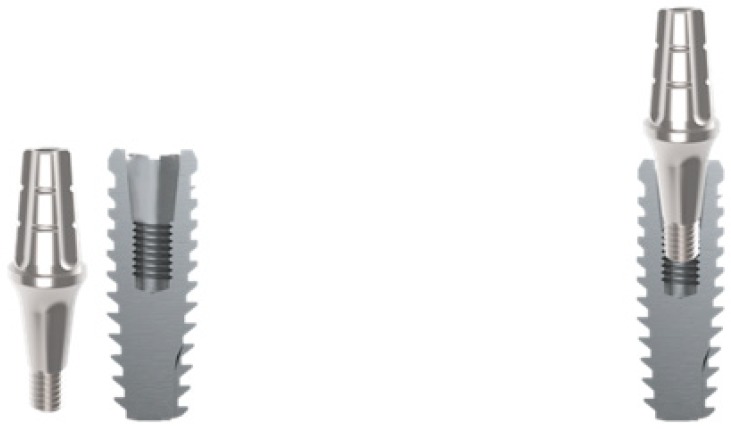
Classification of implant groups according to implant type, implant diameters and lengths, and the region of the prosthetic abutment before the mechanical test.

**Figure 2 materials-13-01888-f002:**
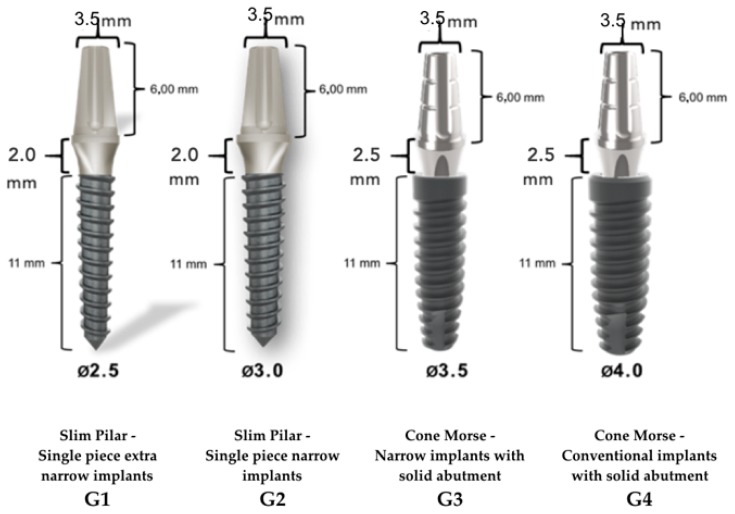
Implacil de Bortoli cone Morse implants, with 6 mm height solid abutment.

**Figure 3 materials-13-01888-f003:**
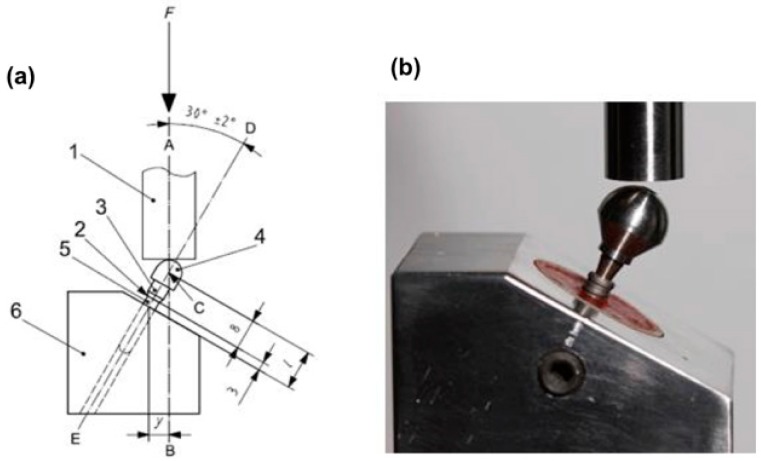
(**a**) Internation Organization for Standardization (ISO) 14801:2007 standard (1. force device; 2. bone level; 3. component; 4. hemispheric loading component; 5. implant body; 6. test body support); (**b**) sample placed on the device to perform the test.

**Figure 4 materials-13-01888-f004:**
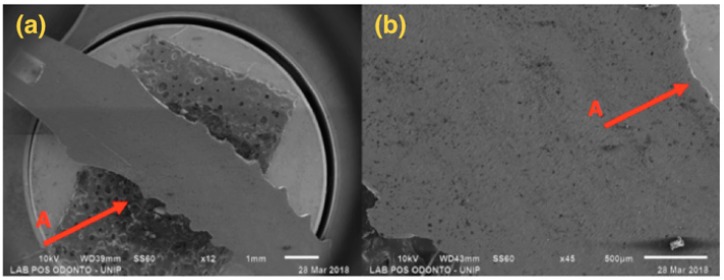
G1: Photomicrography: (**a**) Magnification of 12×; (**b**) Magnification of 45×; A—Distortion is seen in the region of the implant body.

**Figure 5 materials-13-01888-f005:**
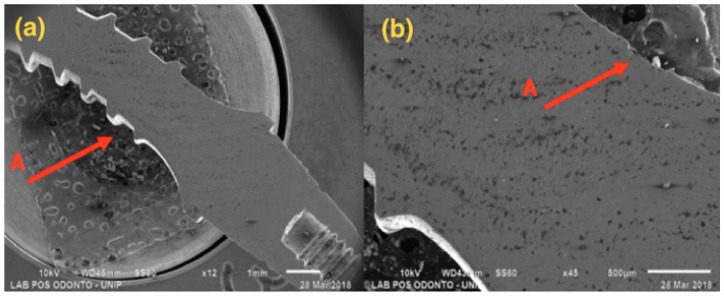
G2: Photomicrography: (**a**) Magnification of 12×; (**b**) Magnification of 45×; A—Distortion is seen in the region of the implant body.

**Figure 6 materials-13-01888-f006:**
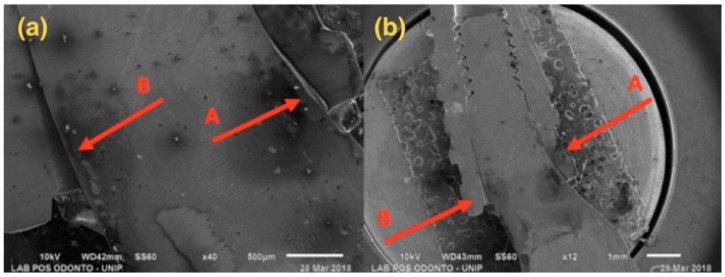
G3: SEM: (**a**) Magnification of 40×; (**b**) Magnification of 12×; A—Compression at the abutment/implant interface; B—Opening at the abutment/implant interface.

**Figure 7 materials-13-01888-f007:**
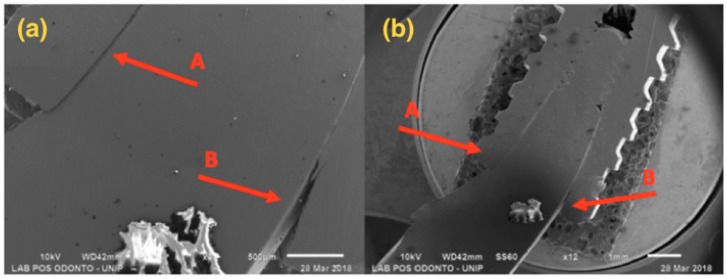
G4: SEM: (**a**) Magnification of 45×; (**b**) Magnification of 12×; A—Compression at the abutment/implant interface; B—Opening at the abutment/implant interface.

**Figure 8 materials-13-01888-f008:**
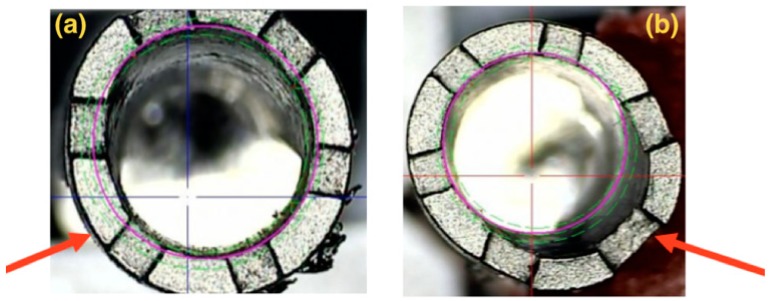
Illustrative images showing the implant superimposition of Groups 3 (**a**) and 4 (**b**) on the profilometer, in which distortion is identified on the platform of each implant after the abutment removal (indicated by the arrows).

**Table 1 materials-13-01888-t001:** Mean and standard deviation values for the maximum force, the tensile strength, and the maximum bending moment of each group. In the columns, different letters (a, b, c, d) indicate statistical difference for the Tukey test (*p* < 0.05).

	Maximum (N)	Tensile Strength (MPa)	Maximum Bending Moment (Nmm)
Group 1	134.29 (10.27) a	22.38 (1.71) a	1316.02 (100.6) a
Group 2	300.61 ( 24.26) b	50.1 (4.04) b	2945.97 ( 237.7) b
Group 3	360.24 (23.34) c	60.04(3.89) c	3530.38 (228.75) c
Group 4	419.1 (18.87) d	69.85(3.14) d	4096.7 (182.73) d

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
