# Peer review of "A Comparative Analysis of Implants Presenting Different Diameters: Extra-Narrow, Narrow and Conventional"

_materials, 2020, doi:10.3390/ma13081888_

Round 1
Reviewer 1 Report
The article is interesting and can be considered for publication
however, an important concern has to be raised. The Authors did not consider some studies published about an innovative double conical connection that reported both clinical than microbiological results. So, it's important to complete the discussion about their result, consider the following publications:
New Microbiol. 2016 Jan;39(1):49-56. Evaluation of resistance against bacterial microleakage of a new conical implant-abutment connection versus conventional connections: an in vitro study.
Int J Environ Res Public Health. 2019 Mar 7;16(5). pii: E829. doi: 10.3390/ijerph16050829. Conventional versus Digital Impressions for Full Arch Screw-Retained Maxillary Rehabilitations: A Randomized Clinical Trial.
Journal of Osseointegration 2017 9(3), pp. 271-275 Immediate versus delayed loading of a new conical connection implant in the esthetic zone: A randomized study with 2-year follow-up
Author Response
Dear Reviewer,
We appreciate all the notes and considerations raised.
The modifications was accepted and made.
We put two of the three references indicated.
Please see the attachment.
Kind Regards,
Alessandra Sayuri Tuzita

Reviewer 2 Report
I suggest the title be changed to “A Comparative Analysis of Implants Presenting Different Diameters: Extra-Narrow, Narrow and Conventional”
Regarding the author's affiliations, there is not a number 6 at authors names
Line 18: remove the word “research”
Lines 30-32: this information should be removed. This is not the study conclusion since this not evaluated. Please change to the study conclusions.
Please put “p” from statistical significance, “per” and “et al.” in italic form
The manuscript needs to be checked regarding the English. Several errors exist that reduce the manuscript quality.
The introduction section needs to be rewritten. The presented information does not follow a coherent sequence. For example, the information presented at lines 47-49 is repeated from that presented in lines 42-43.
Line 66-67: when referring to the study hypothesis as the aim of the study, the word shows cannot be used, since that can only be concluded after the results.
Table 1 caption needs to be improved
How was the number of specimens in each group determined? Did the authors perform a sample calculation test previous to the experimental work?
Table 1 and figure 1 show repeated information, please remove one or remove repeated information
Figure 1: if the authors decide to maintain the figure, please add G1, G2, G3 and G4 bellow each of the implants
Figure 1 caption presents repeated information from the figure, please remove it
Line 98: the authors refer that: “For cone morse implants (Group 3 and Group 4), the abutment was installed with a torque of 30 Newtons, following the manufacturer's recommendations (Figure 2)." However, figure 2 does not show that just represents the implant and the abutment. Please change it.
Lines 131-133 should be moved above, after line 127
Line 152 should be removed, since that information was already presented before, at lines 126-127
Line 151: remove “such as” and replaced by a more appropriate word as “of”
Change figures numbers from figure 4A, 4B, 4C and 4D, to figures 4,5,6 e 7 since they are presented as separated figures and not in a panel
In each figure (4,5,6 and 7), place A and B in the figure, and refer in the figure caption the magnification of A and B
Figure 5 should be renamed as figure 8. Indicate in figure caption what the red arrows mean
Lines 218-220: this information should be moved to the discussion section
Lines 224-226: paragraph should be rewritten since it presents repeated information in two sentences
Line 234: please explain what means: “as they are well reported”
Line 237: phrase should be rewritten since it is confusing “which the mechanical test would fail survival”
Lines 239-230: the authors refer that at group 1 (2.5 mm diameter) the mean value for the maximum force supported was 154.46N. However, in table 2, the maximum value presented was 134.29. What is the correct value?
Lines 294-295: the authors conclude about bone resorption around extra-narrow implants, based on a systematic review but no based on the original results obtained. Please discuss your results and their influence on this topic.
Lines 299-230 need to be removed. The authors did not observe the success in the rehabilitation of lost teeth since this is an in vitro study
Lines 317-319 need to be removed since this study is insufficient to support clinical guidelines
Please add a paragraph regarding the study limitations to the discussion section
The discussion section needs to be rewritten and more focused on discussing the obtained results
Author Response
Dear Reviewer,
We appreciate all the notes and considerations raised.
The modifications was accepted and made.
Please see the attachment.
Kind Regards,
Alessandra Sayuri Tuzita

Reviewer 3 Report
Abstract: your statement concerning the indication of narrow implants (lateral incisors, lower incisors) is not based on your own results.
Methods and Material: The calculation of your maximum force according to ISO 14801 is unclear. You have to specify a critical point of failure, which may be a certain percentage of elastic or permanent deformation, loosening of an abutment or fracture of a component. Figures 4b and 4c already show substantial permanent deformation in the acrylic resin. For better clarification it might be useful to show an actual graph.
In line 125 you mention a table without a number.
Results: There is too little information about the implant behavior under loading. The authors only mention “maximum load” without defining a critical point. For instance initial bending would be an important critical point.
Figures 5a and b show light microscopic images. The headline (Profilometer, line 207) is misleading. Please specify your light microscopic Equipment.
Discussion:
The discussion lacks critical addressing of the methods. Which point of failure did you choose and why.
What might be the influence of the acrylic embedding material, for most of the permanent deformation was within the embedded area. The SEM pictures show a very porous material around the segmented implants. Where the implants removed from the acrylic? Please clarify.
According to the ISO, the 30° angulation simulates a worst cases scenario, please discuss that fact.
Line 228: I think your statement concerning implant failure refers to mechanical failure. Clinically the most common failure is peri-implantitis – please clarify.
Line 232 to 233: it is hard to understand what you mean with your statement “as there are reports only for edentulous areas with no occlusal load” for each functioning implant is exposed to occlusal load clinically.
Line 273 All the information concerning initial bending is crucial to the clinician. However that information is not given in your results
Line 236 to 240 is hard to understand. You state that loads in the incisor region is between 117N and 227 N. Then you state that the maximum force of your 2.5 mm implants was 154.5N.
Please clarify which situation (line 239) has not occurred. I assume you want to state that the 2.5mm implants are to weak to withstand anterior occlusal forces.
Line 264: From the Methods and material section the implants were exposed to a bending test and not to a compression test.
Line 299 to 300: Please provide a reference for that Statement.
Abstract: your statement concerning the indication of narrow implants (lateral incisors, lower incisors) is not based on your own results.
Methods and Material: The calculation of your maximum force according to ISO 14801 is unclear. You have to specify a critical point of failure, which may be a certain percentage of permanent deformation, loosening of an abutment or fracture of a component. Figures 4b and 4c already show substantial permanent deformation in the acrylic resin. For better clarification it might be useful to show an actual graph.
In line 125 you mention a table without a number.
Results: There is too little information about the implant behavior under loading. The authors only mention “maximum load” without defining a critical point. For instance initial bending would be an important critical point.
Figures 5a and b show light microscopic images. The headline (line 207) is misleading.
Discussion:
The discussion lacks critical addressing of the methods. Which point of failure did you choose and why.
What might be the influence of the acrylic embedding material, for most of the permanent deformation was within the embedded area. The SEM pictures show a very porous material around the segmented implants. Where the implants removed from the acrylic? Please clarify.
According to the ISO, the 30° angulation simulates a worst cases scenario, please discuss that fact.
Line 228: I think your statement concerning implant failure refers to mechanical failure. Clinically the most common failure is peri-implantitis – please clarify.
Line 232 to 233: it is hard to understand what you mean with your statement “as there are reports only for edentulous areas with no occlusal load” for each functioning implant is exposed to occlusal load clinically.
Line 273 All the information concerning initial bending is crucial to the clinician. However that information is not given in your results
Line 236 to 240 is hard to understand. You state that loads in the incisor region is between 117N and 227 N. Then you state that the maximum force of your 2.5 mm implants was 154.5N.
Please clarify which situation (line 239) has not occurred. I assume you want to state that the 2.5mm implants are to weak to withstand anterior occlusal forces.
Line 264: From the Methods and material section the implants were exposed to a bending test and not to a compression test.
Line 299 to 300: Please provide a reference for that Statement.
Author Response

(The authors gave the same response as above.)

Round 2
Reviewer 2 Report
Line 48: I do not agree with the sentence: "avoiding bone reconstruction". Narrow implants can be used in situations were no bone reconstruction is needed. I suggest this part be removed.
Lines 41-48 and 55-57 present repeated information. Please merge the sentences and present that information in a logical and sequential way
Line 80: I suggest the expression “and compared according to the figure 1 below” to be changed to: “as shown in figure 1”
Line 102-103: I suggest the phrase to be changed from: “For cone morse implants (Group 3 and Group 4), figure 2 referring that the abutment was installed with a torque of 30 Newtons, following the manufacturer's recommendations.” To “For cone morse implants (Group 3 and Group 4), a 6 mm height solid abutment was inserted (figure 2) with a torque of 30 Newtons, following the manufacturer's recommendations”
Figure 3: please add “A” and “B” to the figure, as described in the figure caption
At the response to reviewers from revision round 1, the authors stated that the sample calculation was determined based on a pilot study. This information should be added to the manuscript.
Figure 6 is repeated, please remove the one without 1 and 2 in the figure
Figure 8: I suggest using 8.1 and 8.2 instead of A and B, to make figures uniform trough the manuscript
Figure 8 caption: I suggest the text to be changed to: “…showing the implant superimposition of groups 3 (8.1) and 4 (8.2) on the profilometer…”
Line 237: since at study aim, the authors removed the reference to the null hypothesis, this information should also be removed here
Line 271: I suggest that "pure titanium" be removed from the sentence: "…which is a pure titanium with…" since this is already referred in the sentence before
Line 299: please put “in vitro” in italic form
Author Response
Dear Reviewer,
We appreciate all the notes and considerations raised.
The modifications was accepted and made.
Please see the attachment.
Kind Regards,
Ms. Alessandra Sayuri Tuzita.

Reviewer 3 Report
The authors did extensive corrections of their manuscript and considered most recommendations and suggestions of the reviewer have been addressed. The paper now is scientifically sound and of interest for the scientific community. Thus the reviewer recommends to publish the paper “as is” in the present form.
Author Response
Dear reviewer, thanks for consideration.
Ms. Alessandra Sayuri Tuzita.